# Large-scale identification of plasma membrane repair proteins revealed spatiotemporal cellular responses to plasma membrane damage

**Yuta Yamazaki, Keiko Kono***

Okinawa Institute of Science and Technology Graduate University, Okinawa, Japan

## eLife Assessment

This work provides an **important** resource identifying 72 proteins as novel candidates for plasma membrane and/or cell wall damage repair in budding yeast, and describes the temporal coordination of exocytosis and endocytosis during the repair process. The data are **convincing**; however, additional experimental validation will better support the claim that repair proteins shuttle between the bud tip and the damage site.

**\*For correspondence:**
keiko.kono@oist.jp

**Competing interest:** The authors declare that no competing interests exist.

**Abstract** Damage to the plasma membrane (PM) is common in all types of cells. PM repair processes, including exocytosis and endocytosis, are not mutually exclusive; rather, they collaborate to repair the wound. However, the temporal coordination between the repair processes remains poorly understood. Here, by large-scale identification and live-cell imaging of PM repair proteins, we analyzed the spatiotemporal PM damage responses in *Saccharomyces cerevisiae*. Of the 80 repair proteins identified, 72 proteins were previously unreported repair protein candidates. Among the observed repair processes, the polarized exocytosis and clathrin-mediated endocytosis (CME) are coupled at the damage site, with exocytosis predominating in the early stage of PM repair and CME predominating in the late stage of PM repair. Furthermore, we showed that CME at the growing bud site directs PM repair proteins with transmembrane domains to the damage site. We propose a model in which CME delivers repair proteins with transmembrane domains between the growing bud site and the damage site. This study provides a functional catalog of PM repair proteins and insights into spatiotemporal cellular responses to PM damage.

## Introduction

The plasma membrane (PM) functions as a defensive barrier in all types of cells to protect cellular components from extracellular stimuli. PM frequently experiences physiological and pathological damages, such as eccentric contraction-induced injuries, migration-induced injuries, pore formation by bacterial toxins, or invasion by cancer cells (*McNeil and Kirchhausen, 2005*; *Dias and Nylandsted, 2021*). Cells undergo cell lysis if the damaged PM is not repaired. Deficits in PM repair are linked to multiple diseases, including limb-girdle muscular dystrophy (*Bashir et al., 1998*) and Scott syndrome (*Suzuki et al., 2010*; *Wu et al., 2020*). Therefore, virtually all cells have PM repair machinery.

In eukaryotes, the influx of $Ca^{2+}$ triggers PM repair (*McNeil and Kirchhausen, 2005*). Upon $Ca^{2+}$ influx, fundamental cellular mechanisms, including exocytosis (*Bittel and Jaiswal, 2023*; *Raj et al., 2023*; *Reddy et al., 2001*; *Bi et al., 1995*; *Miyake and McNeil, 1995*), endocytosis (*Idone et al., 2008*; *Tam et al., 2010*), membrane shedding by ESCRT complex (*Jimenez et al., 2014*; *Scheffer*

*et al., 2014*), and the constriction forces by actin cytoskeleton (*Mandato and Bement, 2003*; *Bement et al., 2005*; *Abreu-Blanco et al., 2011*; *Benink and Bement, 2005*) are directed to the damage site to reseal the wound. These initial repair processes usually occur within 1 min after PM damage (*Ebstrup et al., 2021*). After the resealing of the damaged PM, cells restructure the damaged PM (*Ebstrup et al., 2021*; *Sønder et al., 2021*; *Raj et al., 2024*). Restructuring is defined as the process by which cells modify the damaged PM to restore PM homeostasis and normal cell function (*Ebstrup et al., 2021*; *Sønder et al., 2021*; *Raj et al., 2024*). These repair processes are not mutually exclusive, and they collaborate to repair the damaged PM (*Ammendolia et al., 2021*). However, the temporal coordination between the PM repair processes remains poorly understood.

Both resealing and restructuring of the damaged PM are mediated by the proteins, which accumulate at the damage site. Here, we defined them as PM repair proteins. PM repair proteins include ESCRT, annexins, and dysferlin in mammalian cells (*Jimenez et al., 2014*; *Lennon et al., 2003*; *Lauritzen et al., 2015*; *McDade et al., 2014*). Using budding yeast *Saccharomyces cerevisiae* as a model, we previously identified evolutionarily conserved repair proteins, including protein kinase C, exocyst, and phospholipid flippases (*Kono et al., 2012*; *Yamazaki and Kono, 2022*). These repair proteins accumulate at the damage site at the appropriate time and proceed with the repair processes. Therefore, identifying repair proteins and observing their movements after PM damage is an important step toward understanding the temporal PM repair processes.

Here, we performed proteome-scale screening of PM repair proteins using yeast GFP-tagged libraries and laser-induced injury assay in budding yeast. We identified 80 repair protein candidates, including 72 previously unreported candidates. Among the observed repair processes, we showed that polarized exocytosis and clathrin-mediated endocytosis (CME) are coupled at the damage site, with exocytosis predominating in the early stage of PM repair and CME predominating in the late stage of PM repair. We also showed that CME delivers repair proteins with transmembrane domains (TMDs) from the growing bud site (bud tip) to the damage site, presumably contributing to the restructuring of the damaged PM. These results provide insights into the temporal dynamics of coordinated cellular responses to PM damage.

## Results

### Proteome-scale identification of SDS-responsive proteins

To identify the PM repair proteins, we performed a two-step screening. First, we aimed to identify the proteins that change localization in response to SDS treatment, which induces local PM/cell wall damage to budding yeast (*Suda et al., 2024*). We used a yeast C-terminally GFP-tagged library (4159 ORFs) (*Huh et al., 2003*) and an N-terminally sfGFP-tagged library (N' Swat library) (5569 ORFs) (*Yofe et al., 2016*; *Weill et al., 2018*) comprising 86% of the yeast proteome (5718 ORFs in total) (*Figure 1A*). We fixed the cells with paraformaldehyde after 1 hr of SDS treatment, in which condition one of the known repair proteins, Pkc1-GFP, changes its localization (*Figure 1A*). We successfully imaged the signal of 9181 proteins fused with GFP or sfGFP (5609 ORFs in total). We assessed the fluorescence signals of 5609 proteins in normal and SDS-treated conditions by visual inspection of the images. We identified 562 proteins whose fluorescence signal pattern changed after SDS treatment (*Figure 1B* and *Figure 1—source data 1*). The hits included the localization changes of proteins, structural changes of organelles such as mitochondrial fragmentation, and foci/puncta formation in response to SDS treatment. In addition to Pkc1, we identified previously reported repair proteins, such as Rom2 and Dnf1, as screening hits (*Kono et al., 2012*; *Yamazaki and Kono, 2022*; *Figure 1—source data 1*). Gene Ontology (GO) analysis of the screening hits revealed enrichment for proteins involved in the actin filament organization (*Figure 1C*). This is consistent with the previous study that SDS treatment remodels the cell polarity and actin cytoskeleton (*Kono et al., 2012*). Based on the reported subcellular localization of proteins in normal growth conditions (*Huh et al., 2003*), we categorized the screening hits into six major classes and several minor ones representing less than 10 proteins (*Figure 1D*, *Figure 1—figure supplement 1*, see Materials and methods). We found that proteins that form puncta/foci after the SDS treatment include translation factor Gcd7-GFP (*Figure 1D*), which forms foci in response to glucose deprivation (*Moon and Parker, 2018*). These observations suggest that SDS treatment induces stress responses, including those associated with PM/cell wall damage responses in budding yeast.

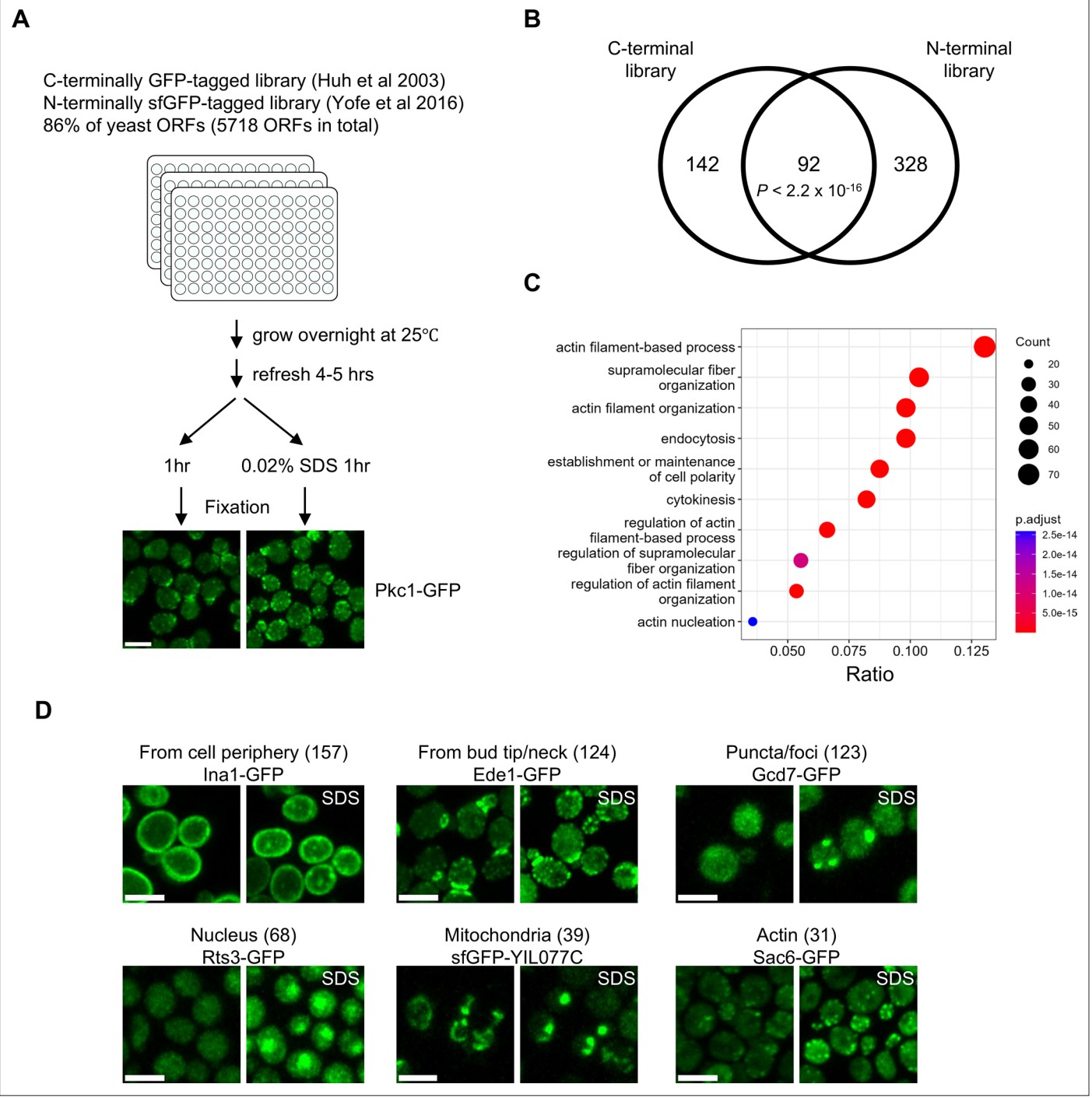

**Figure 1.** Protein relocalization in response to SDS treatment. (**A**) Schematic representation of screening methodology and images of Pkc1-GFP with or without 0.02% SDS treatment are shown. Scale bar, 5 μm. (**B**) Overlap of screening hits of C- and N-terminal libraries. p-Value for the significance of the overlap is indicated. Fisher's exact test was performed. (**C**) Gene Ontology (GO) analysis of biological processes for the screening hits. Successfully observed proteins were used as the background protein sets. (**D**) Screening hits for six relocalization classes and the images of representative proteins in each class were shown. Numbers in parentheses indicate the number of proteins in the class. Scale bar, 2 μm.

The online version of this article includes the following source data and figure supplement(s) for figure 1:

**Source data 1.** Proteins whose localization changes in response to the SDS treatment.

**Figure supplement 1.** Minor classes of localization changes in response to SDS treatment.

## Laser-induced PM/cell wall damage assay identified 80 repair protein candidates

Our group previously developed a laser-induced PM/cell wall injury assay under live single-cell conditions (laser damage assay) (*Kono et al., 2012*; *Kono et al., 2016*). We performed the laser damage assay to identify the PM repair proteins and observe their movements after PM/cell wall damage (*Figure 2A*). The N-terminally sfGFP-tagged library is expressed under exogenous *NOP1* promoters, while the C-terminally GFP-tagged library is expressed under endogenous promoters (*Huh et al., 2003*; *Yofe et al., 2016*). To assess the proteins with endogenous expression levels, we selected the 234 screening hits from the C-terminal library as targets for the laser damage assay. During 25 min of observation at 30 s intervals, we identified 90 proteins whose localization changed in response to laser damage (*Figure 2B*). We categorized the proteins into four classes based on their localization changes (*Figure 2C*). Three of the puncta-forming proteins, Dna2, Dot6, and Gcd7, form puncta in response to cellular stresses (*Moon and Parker, 2018*; *Tkach et al., 2012*). The internalization of PM proteins (Dip5, Ina1, Ftr1, and Tpo1) is consistent with the fact that endocytosis of PM proteins contributes to cell adaptation to environmental stress (*López-Hernández et al., 2020*). In addition, the transcription factors, Msn2 and Crz1, translocate from the cytoplasm to the nucleus after laser damage (*Figure 2B*). Msn2 and Crz1 are activated under various stress conditions (*Camponeschi et al., 2023*). Thus, the localization changes of puncta-forming proteins and transcriptional factors suggest common stress responses to laser damage and other cellular stresses. The most frequently observed localization changes were the accumulation at the damage site (*Figure 2B*). We defined them as repair protein candidates, given their presumed involvement in the PM/cell wall repair process.

To gain further insight into the repair protein candidates, we classified them into four categories based on their subcellular localization under normal growth conditions (*Figure 2C*). The largest category is bud-localized proteins (47 proteins) (*Figure 2C*), which is consistent with a previous study showing that bud-localized proteins accumulate at the damage site in budding yeast (*Kono et al., 2012*). The repair protein candidates in this category include 10 proteins that have TMDs, including phospholipid flippases (Dnf1/Dnf2) and osmosensor proteins (Slg1, Sho1) (*Figure 2C*). Proteins with TMDs are transported to the PM as cargoes of secretory vesicles (*Shimizu and Uemura, 2022*). Thus, this result suggests that secretory vesicles targeted to the damage site provide repair proteins with TMDs to the damage site. The second largest category is actin-localized proteins (25 proteins). This is consistent with the GO analysis that actin-binding proteins and proteins involved in actin-related biological processes are enriched in the repair protein candidates (*Figure 2D and E*). The proteins in this category include endocytic proteins, suggesting that endocytosis occurs at the damage site. Proteins that bind actin cables, such as Abp140, are included in this category, suggesting that actin cables are formed at the damage site. In addition, proteins involved in cell wall synthesis/maintenance (Flc1, Dfg5, Smi1, Skg1, Tos7, and Chs3) are identified as repair protein candidates (*Figure 2—source data 1*). These results are consistent with previous studies of cell wall damage repair in budding yeast and PM repair in both yeast and human cells (*Idone et al., 2008*; *Kono et al., 2012*; *Levin, 2005*; *Levin, 2011*).

## The temporal order of the Pkc1 accumulation, polarized exocytosis, and CME at the damage site

To understand when repair proteins accumulate at the damage site, we defined their accumulation times as the time when the fluorescence intensity at the damage site exceeded the threefold standard deviation (3×SD) above the non-damaged site for at least two consecutive time frames (1 min). We also defined dispersion time when the fluorescence intensity at the damage site becomes less than that of the non-damaged site plus 3×SD for at least 1 min (*Figure 2—source data 1*).

The defined accumulation times raised the possibility that the Pkc1 pathway proteins accumulated at the damage site earlier than exocytosis regulators and exocytic cargoes with TMDs (Dnf1, Dnf2, and Slg1) (*Figure 3—figure supplement 1*). To verify this observation, we performed the laser damage assay in the cells expressing Exo70-mNeonGreen (mNG) and Pkc1-mScarlet-I (mSc-I), and the cells expressing Pkc1-sfGFP and Dnf1-mSc-I (*Figure 3A and B*). Exo70 is one of the subunits of the exocyst complex, serving as a polarized exocytosis marker. The fluorescence-tagged Pkc1 accumulated at the damage site earlier than Exo70-mNG and Dnf1-mSc-I (*Figure 3A and B*). These results

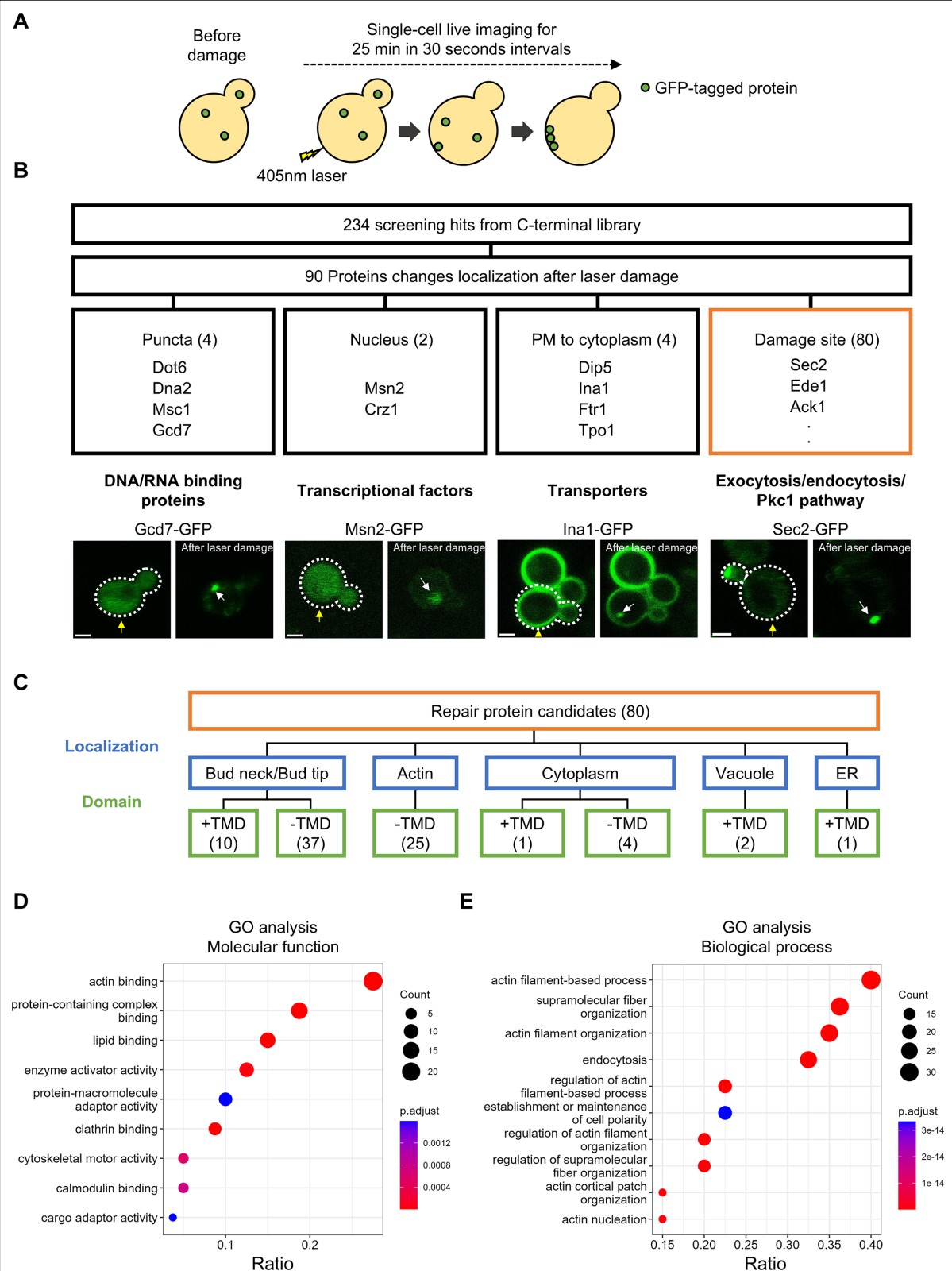

**Figure 2.** Laser damage assays identified 80 repair protein candidates. (**A**) Schematic representation of screening methodology. Cells were imaged for 25 min in 30 s intervals after 405 nm laser damage. (**B**) 90 proteins change localization after laser damage. The classification of the repair protein candidates based on their localization changes, representative proteins, and representative biological processes in each category is shown. Scale bar, 2 μm. Yellow arrows show the damage site. White arrows show the recruitment of fluorescence signals. (**C**) Classification of repair protein candidates

*Figure 2 continued on next page*

*Figure 2 continued*

based on subcellular localization and domain. TMD+ represents the proteins that have transmembrane domains. The existence of transmembrane domains was predicted by TMHMM (*Krogh et al., 2001*). (**D**) Gene Ontology (GO) analysis of molecular functions for the repair protein candidates. The proteins whose localization changes in response to SDS treatment were used as background protein sets. (**E**) GO analysis of biological processes for the repair protein candidates. The proteins whose localization changes in response to SDS treatment were used as background protein sets. The ratio of the number of proteins associated with a specific GO term to the total number of proteins in the background is shown.

The online version of this article includes the following source data for figure 2:

**Source data 1.** Repair protein candidates.

**Source data 2.** Quantification data of fluorescence signal of repair proteins.

suggest that the Pkc1 accumulation at the damage site occurs earlier than the polarized exocytosis of transmembrane proteins.

Although not all exocyst subunits (eight subunits) were identified in this work, all the mSc-I-tagged exocyst subunits showed comparable fluorescence intensity changes with Exo70-mNG at the damage site and at the bud tip (*Figure 3—figure supplement 2A–G*). These results suggest that all subunits of the exocyst complex accumulate at the damage site simultaneously.

The CME markers, Sla1-GFP and Abp1-GFP (*Kaksonen et al., 2005*), exhibited repeated short stays and reaccumulation at the damage site within 5 to 25 min after laser damage (*Figure 3—figure supplement 3A–C*). These fluctuating movements of Sla1 and Abp1 are consistent with the previous studies (*Kishimoto et al., 2011*; *Sun et al., 2015*). Furthermore, Sla1-mNG and Abp1-mSc-I showed comparable fluctuating accumulation patterns in the same cell (*Figure 3—figure supplement 3D*). These results suggest that CME repeatedly occurs at the damage site from around 5 to 25 min after laser damage.

To understand the temporal order of polarized exocytosis and CME at the damage site, we performed the laser damage assay in the cells expressing Exo70-mNG/Dnf1-mNG and Ede1-mSc-I (*Figure 3C and D*). Ede1 is one of the earliest proteins recruited to endocytic sites, serving as a CME marker (*Lu and Drubin, 2017*; *Stradalova et al., 2012*). Exo70-mNG/Dnf1-mNG and Ede1-mSc-I accumulated at the damage site from 5 to 35 min (*Figure 3C and D*). The signals of Exo70-mNG/Dnf1-mNG peak within 20 min after the damage, while Ede1-mSc-I peaks 20 min after the damage (*Figure 3C and D*). These results suggest that CME and polarized exocytosis occur simultaneously at the damage site, but that polarized exocytosis predominates in the early stage of the PM/cell wall damage response, while CME predominates in the late stage of the PM/cell wall damage response.

## Knockout mutants of CME are sensitive to PM/cell wall stresses

To identify the biological processes required for cell survival after PM/cell wall stress, we performed growth screening of repair protein knockout mutants in PM/cell wall stress conditions (*Figure 4—figure supplement 1A*). We spotted the same number of yeast cells in YPD media, YPD+0.01% SDS media, YPD+25 µg/ml calcofluor white (CFW), and YPD media in 37°C (heat stress) and incubated for 3 days (*Figure 4—figure supplement 1A*). CFW binds to chitin in the cell wall, inducing cell wall damage (*Ram et al., 1994*). Heat stress modifies PM structures (*Török et al., 2014*). The growth assay showed that six knockout mutants of CME proteins, *rvs167Δ*, *sla1Δ*, *sla2Δ*, *end3Δ*, *las17Δ*, and *vrp1Δ*, are sensitive to all the stress conditions (*Figure 4—figure supplement 1B*). We knocked out these genes using the WT strain in our laboratory and confirmed that these mutants are sensitive to SDS (*Figure 4—figure supplement 1C*). These results led us to focus further on the CME functions associated with PM/cell wall damage repair.

## CME proteins are required for polarized exocytosis at the damage site

Previous studies showed that endocytosis is involved in PM damage repair by directly resealing the PM by removing the damaged pores (*Idone et al., 2008*) or by restructuring the PM after resealing in mammalian cells (*Sønder et al., 2021*; *Raj et al., 2024*; *Skalman et al., 2018*). We showed that CME occurs at the damage site from 5 to 45 min after laser damage in budding yeast (*Figure 3B and C*). This supports the possibility that CME restructures the PM after resealing, which usually occurs within 1 min after the damage (*Sønder et al., 2021*). Another potential mechanism for restructuring the damaged PM is exocytosis, which delivers PM proteins and lipids to the damage site. Our previous

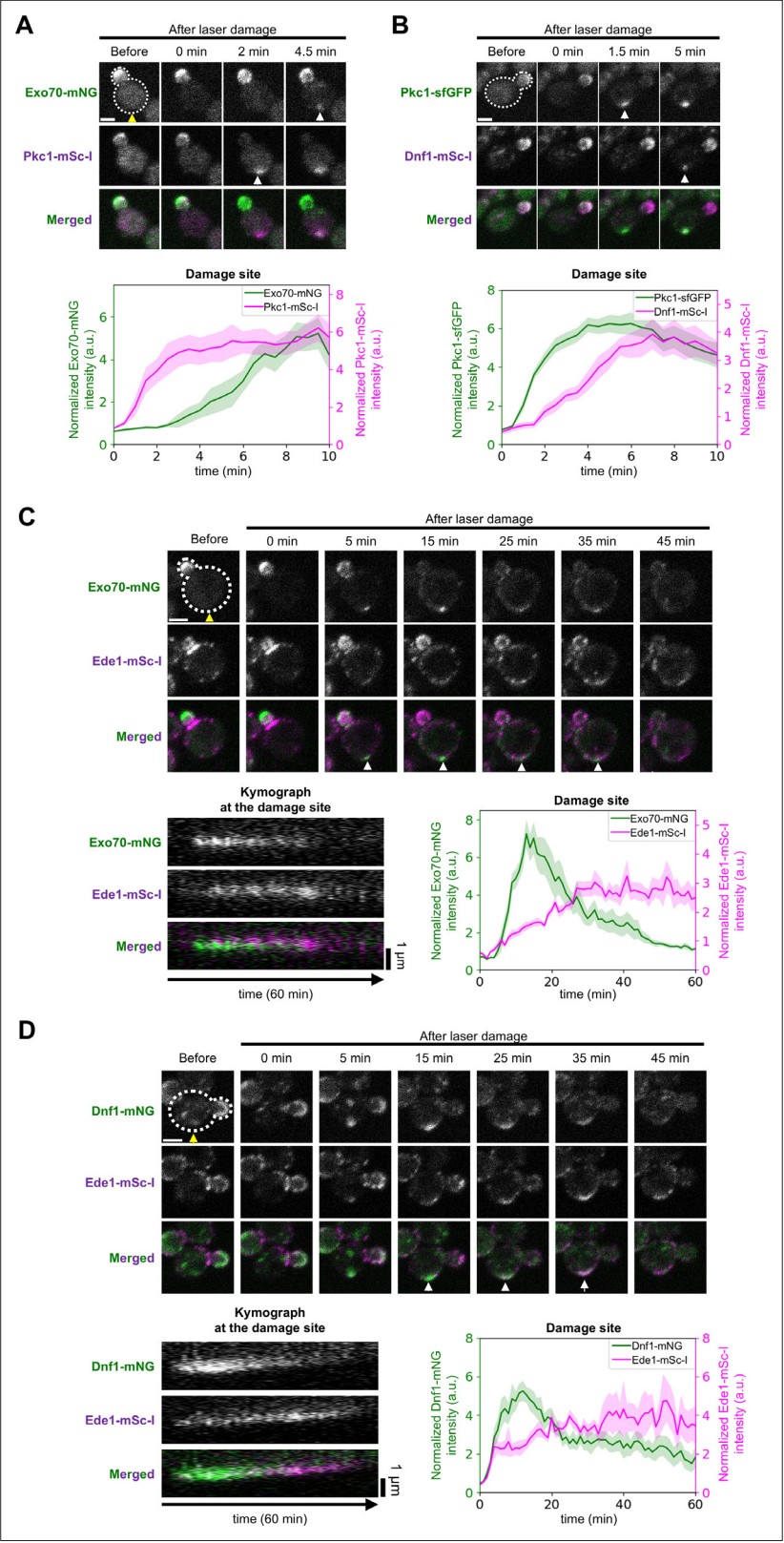

**Figure 3.** The temporal order of the Pkc1 accumulation, polarized exocytosis, and clathrin-mediated endocytosis (CME) at the damage site. (**A**) The representative images and normalized fluorescence intensity of Exo70-mNG (green) and Pkc1-mSc-I (purple) at the damage site after laser damage. Yellow arrows show the damage site. White arrows show the recruitment of fluorescence signals. n=10 cells. (**B**) The representative images and normalized

*Figure 3 continued on next page*

*Figure 3 continued*

fluorescence intensity of Pkc1-sfGFP (green) and Dnf1-mSc-I (purple) at the damage site after laser damage. Yellow arrows show the damage site. White arrows show the recruitment of fluorescence signals. n=8 cells. (**C**) Representative images, kymograph at the damage site, and fluorescence intensity at the damage site of Exo70-mNG (green) and Ede1-mSc-I (purple). n=10 cells. Yellow arrows show the damage site. White arrows show the coaccumulation of Exo70-mNG and Ede1-mSc-I at the damage site. (**D**) Representative images, kymograph at the damage site, and fluorescence intensity at the damage site of Dnf1-mNG (green) and Ede1-mSc-I (purple). n=9 cells. Yellow arrows show the damage site. White arrows show the coaccumulation of Dnf1-mNG and Ede1-mSc-I. Lines and shaded regions are the mean and the standard error of the mean, respectively. White scale bar, 2 μm.

The online version of this article includes the following source data and figure supplement(s) for figure 3:

**Source data 1.** Quantification data of fluorescence signal of respective repair proteins.

**Figure supplement 1.** Summary of the accumulation time of repair proteins.

**Figure supplement 2.** Accumulation patterns of exocyst subunits.

**Figure supplement 2—source data 1.** Quantification data of fluorescence signal of exocyst subunits.

**Figure supplement 3.** Clathrin-mediated endocytosis (CME) continuously occurs at the damage site.

**Figure supplement 3—source data 1.** Quantification data of fluorescence signal of Sla1-GFP, Abp1-GFP, Sla1-mNG, Abp1-mSc-I.

work showed that the polarized exocytosis machinery is directed from the bud tip to the damage site in response to laser damage in budding yeast (*Kono et al., 2012*). Given that CME and polarized exocytosis constitute a coupled transport cycle in budding yeast (*Johansen et al., 2016*), we reasoned that CME positively regulates polarized exocytosis at the damage site, thereby restructuring the damaged PM.

To test this idea, we performed the laser damage assay in SDS-sensitive endocytic mutants (*sla1Δ*, *end3Δ*, *vrp1Δ*, and *rvs167Δ*). We used two exocytosis markers, type V myosin (Myo2) and the exocyst subunit (Exo70), fused to sfGFP and mNG, to assess the exocytosis activity at the damage site. The effect of deleting the CME genes on the expression levels of Myo2-sfGFP and Exo70-mNG before laser damage was minimal (*Figure 4—figure supplement 2A and B*). The accumulation of Myo2-sfGFP and Exo70-mNG at the damage site was impaired in *sla1Δ*, *end3Δ*, and *vrp1Δ* (*Figure 4A and B*, *Figure 4—figure supplement 2A and B*). Moreover, we also found that the dispersion of Myo2-sfGFP and Exo70-mNG from the bud tip was partially inhibited in CME mutants (*Figure 4A and B*, *Figure 4—figure supplement 2A and B*). The observed weak phenotype of *rvs167Δ* for the accumulation and dispersion of Myo2-sfGFP and Exo70-mNG is consistent with the previous study that exocytosis is not impaired in *rvs167Δ* (*Johansen et al., 2016*).

Our previous work also showed that the actin nucleator formin Bni1 and the exocyst subunit Sec3 are degraded in response to SDS treatment (*Kono et al., 2012*). To test the possibility that CME is required for the decrease of Bni1 and Sec3 after SDS treatment, we performed immunoblotting of Bni1-13xMyc and Sec3-GFP before and after SDS treatment (*Figure 4—figure supplement 3A and B*). Protein levels of Bni1-13xMyc and Sec3-GFP were decreased in WT, *rvs167Δ*, and *end3Δ* after SDS treatment (*Figure 4—figure supplement 3A and B*). Given that *end3Δ* impaired the accumulation of Myo2-sfGFP and Exo70-mNG at the damage site (*Figure 4A and B*), these results suggest that CME is dispensable for the decrease of Bni1 and Sec3 after SDS treatment. Altogether, these results suggest that CME is involved in the direction of the Exo70-mNG and Myo2-sfGFP to the damage site in a Bni1- and Sec3-degradation-independent manner.

## CME at the bud tip directs repair proteins with TMDs to the damage site

Although the requirement of Rvs167 for the Pkc1 accumulation and polarized exocytosis at the damage site is minimal (*Figure 4A–C*), *rvs167Δ* is sensitive to SDS (*Figure 4—figure supplement 1C*). This raised the possibility that CME plays additional roles, such as membrane protein trafficking, in PM repair. We found that CME proteins, including Apl1, Ede1, and Ent1, changed their localization from the bud neck to the bud tip within 10 min after laser damage (*Figure 5—figure supplement 1A*). The fluorescence intensity changes of CME proteins at the bud tip at 10 min after laser damage were

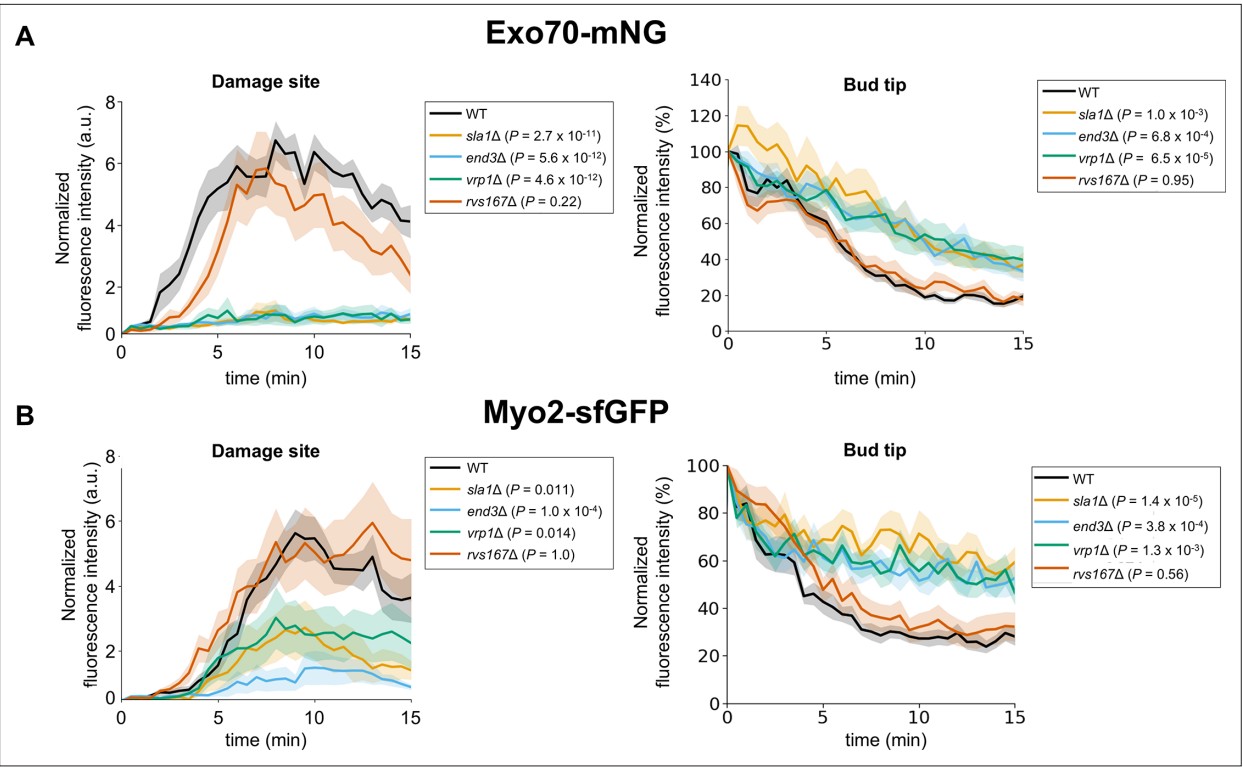

**Figure 4.** Clathrin-mediated endocytosis (CME) proteins are required for polarized exocytosis at the damage site. (**A**) The quantification results of fluorescence intensity of Myo2-sfGFP at the damage site and at the bud tip. n=13 for WT, n=12 for *end3Δ*, *sla1Δ*, *rvs167Δ*, and *vrp1Δ*. (**B**) The quantification results of the fluorescence intensity of Exo70-mNG at the damage site. n=18 for WT, n=12 for *end3Δ*, *sla1Δ*, and *rvs167Δ*, n=14 for *vrp1Δ*. Lines and shaded regions are the mean and standard error of the mean, respectively. The maximum fluorescence intensity at the damage site or fluorescence intensity changes at the bud tip were compared between the WT and each CME mutant using Dunnett's multiple comparison test.

The online version of this article includes the following source data and figure supplement(s) for figure 4:

**Source data 1.** Quantification data of fluorescence signal of Pkc1-sfGFP and Exo70-mNG.

**Figure supplement 1.** Growth screening of repair protein knockout mutants in plasma membrane (PM)/cell wall damage conditions.

**Figure supplement 2.** Representative images of Myo2-sfGFP and Exo70-mNG in clathrin-mediated endocytosis (CME) mutants.

**Figure supplement 2—source data 1.** Quantification data of fluorescence signal of Pkc1-sfGFP and Exo70-mNG before laser damage.

**Figure supplement 3.** Clathrin-mediated endocytosis (CME) proteins are not required for the degradation of Bni1-13xMyc and Sec3-GFP after SDS treatment.

**Figure supplement 3—source data 1.** Original files of western blots.

**Figure supplement 3—source data 2.** PDF files of western blots with sample labels.

higher than those of other repair protein candidates (*Figure 5—figure supplement 1B*). Consistent with these results, the Ede1-mSc-I signal at the bud tip increased within 10 min after laser damage, and then the Ede1-mSc-I signal at the damage site increased (*Figure 5A*). These results suggest that CME occurs at the bud tip within 10 min after laser damage. We also found that Dnf1-mNG disappeared from the bud tip following the recruitment of Ede1-mSc-I to the bud tip (*Figure 5A and B*). The disappearance from the bud tip and the accumulation at the damage site of Dnf1-mNG were impaired in *rvs167Δ* (*Figure 5B and C* and *Figure 5—figure supplement 2*). Furthermore, the disappearance from the bud tip and the accumulation at the damage site of repair proteins with TMDs (Slg1-sfGFP and Sho1-GFP) and representative endocytic recycling cargo, mNG-Snc1, were impaired in *rvs167Δ* (*Figure 5D–F* and *Figure 5—figure supplement 3A–C*). These results support our idea that CME directs repair proteins with TMDs from the bud tip to the damage site.

## mNG-Snc1 is retargeted from the damage site to the bud tip

The fluorescence intensity of accumulated repair proteins with TMDs at the damage site gradually decreases approximately 15 min after laser damage (*Figure 3D*, *Figure 5—figure supplement 3A*

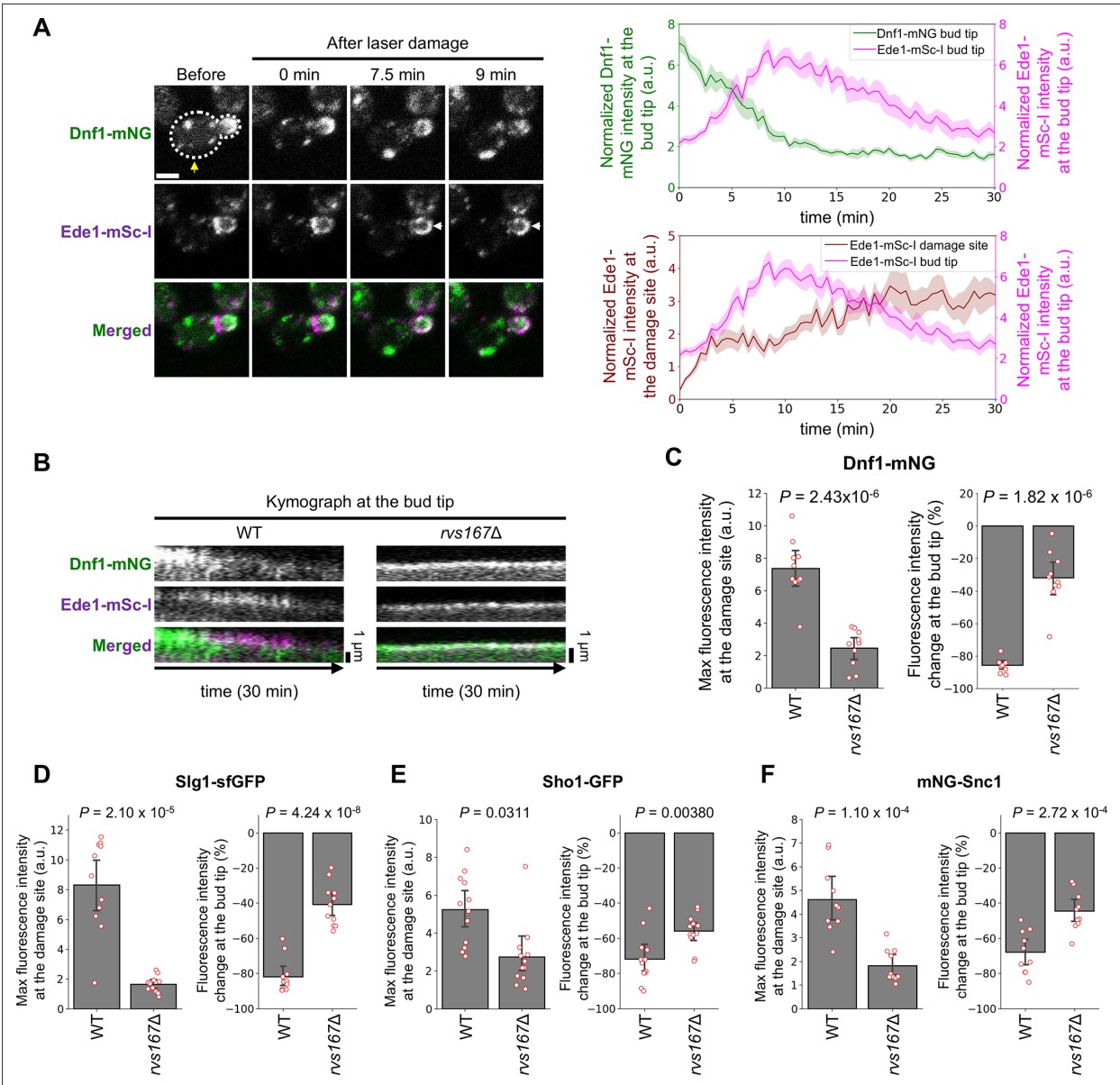

**Figure 5.** Clathrin-mediated endocytosis (CME) at the bud tip directs repair proteins with transmembrane domains (TMDs) to the damage site. (**A**) Representative images and the normalized fluorescence intensity at the bud tip of Dnf1-mNG and Ede1-mSc-I. Yellow arrows show the damage site. White arrows show the recruitment of fluorescence signals. Scale bar, 2 μm. (**B**) Kymograph of Dnf1-mNG and Ede1-mSc-I at the bud tip in WT and *rvs167Δ*. n=10 cells. (**C–F**) Max fluorescence intensity at the damage site and fluorescence intensity changes at the bud tip in WT and *rvs167Δ*. n=10 cells for Dnf1-mNG. n=11 cells for Slg1-sfGFP. n=12 cells for Sho1-GFP. n=10 cells for mNG-Snc1. Welch's t-test was performed.

The online version of this article includes the following source data and figure supplement(s) for figure 5:

**Source data 1.** Quantification data of fluorescence signal of repair proteins at the bud tip.

**Figure supplement 1.** Fluorescence intensity changes of clathrin-mediated endocytosis (CME) proteins at the bud tip.

**Figure supplement 2.** Time course fluorescence intensity changes of Dnf1-mNG and Ede1-mSc-I at the bud tip in *rvs167Δ*.

**Figure supplement 3.** Time course fluorescence intensity changes of repair proteins with transmembrane domains (TMDs).

*and C*). Using mNG-Snc1 as a model, we investigated the destination of repair proteins after PM repair. To avoid observing the mNG-Snc1 that is newly synthesized after laser damage, we transiently expressed mNG-Snc1 under the control of the galactose-inducible promoter (Gal1pr) before laser damage (*Figure 6A*). After 1 hr of mNG-Snc1 expression in galactose media, we transferred the cells to glucose media to stop the expression (*Figure 6A*). To minimize the effect of changing the

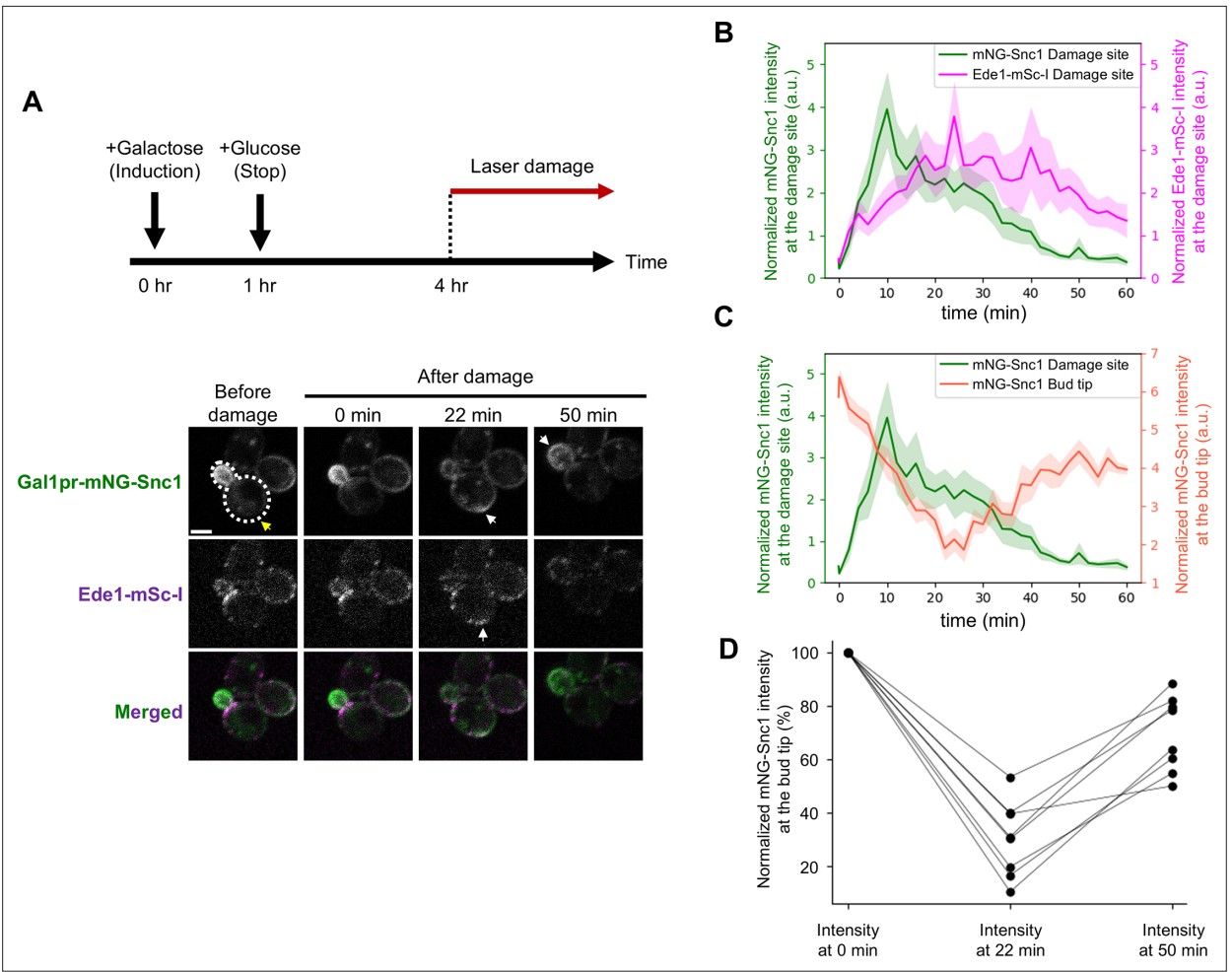

**Figure 6.** mNG-Snc1 is recovered from the damage site to the bud tip after plasma membrane (PM) repair. (**A**) Schematic of transient expression induction of mNG-Snc1 by Gal1 promoter and representative images of mNG-Snc1 and Ede1-mSc-I. After transcription activation of the Gal1 promoter by adding 3% galactose, we stop the expression by transferring the cells to glucose media. The cells were incubated for at least 3 hr before the laser damage assay. Yellow arrows show the damage site. White arrows show the recruitment of fluorescence signals. Scale bar, 2 μm. (**B**) Quantification of mNG-Snc1 (green) and Ede1-mSc-I (purple) at the damage site. (**C**) Quantification of mNG-Snc1 at the bud tip (green) and at the damage site (tomato). (**D**) The changes in the normalized mNG-Snc1 signal at the bud tip. n=8 cells.

The online version of this article includes the following source data for figure 6:

**Source data 1.** Quantification data of fluorescence signal of mNG-Snc1 and Ede1-mSc-I.

---

carbon source, we further incubated the cells in glucose media for at least 3 hr prior to laser damage (***Figure 6A***). The mNG-Snc1 accumulated at the damage site approximately ~10 min after laser damage, gradually disappearing after the colocalization with Ede1-mSc-I (***Figure 6A and B***). The mNG-Snc1 signal at the bud tip decreased to 10–53% 22 min after laser damage (***Figure 6C and D***). The normalized mNG-Snc1 signal at the bud tip recovered to 54–88% 50 min after laser damage (***Figure 6C and D***). These results suggest that mNG-Snc1 is redirected to the bud tip after PM repair.

## Discussion

There has been a growing interest in the mechanisms underlying PM repair, partly due to their association with human diseases and cellular aging (***Bashir et al., 1998***; ***Suzuki et al., 2010***; ***Wu et al., 2020***; ***Suda et al., 2024***). PM repair proteins, which accumulate at the damage site, play critical roles in PM repair. In this study, by large-scale identification of PM repair proteins and single- and dual-color live-cell imaging of repair proteins, we analyzed spatiotemporal PM repair processes in budding yeast. We propose a model in which CME at the bud tip and at the damage site delivers repair proteins with

TMDs between the bud tip and the damage site, allowing the cell to restructure the damaged PM and to resume growth after PM repair (*Figure 7*).

## Two-step visual screening for PM repair protein identification

By combining proteome-scale visual screening using yeast GFP libraries and the laser damage assay, we identified 80 repair protein candidates (*Figure 2B* and *Figure 2—source data 1*). Strikingly, 72 out of 80 repair protein candidates were not previously reported to accumulate at the damage site in budding yeast (*Kono et al., 2012*; *Yamazaki and Kono, 2022*). The unreported repair protein candidates include uncharacterized proteins, such as Sap1 and Ypr089w (*Figure 2—source data 1*). Characterizing these proteins may expand our understanding of PM repair mechanisms.

We selected the screening hits from the C-terminally GFP-tagged library as targets for the laser damage assay. Although this is a reasonable approach to evaluating the proteins from the endogenous expression level, it overlooks the potential repair proteins in the N-terminally sfGFP-tagged library. Around 23% of ORFs only exist in the N-terminal library, and 11% of yeast ORFs show different localization from that of the C-terminal library (*Weill et al., 2018*). For example, sfGFP-Snc2, but not Snc2-GFP, changes localization in response to SDS treatment because of the mislocalization of Snc2-GFP in the vacuole. Because Snc2 is a homolog of Snc1, Snc2 is a potential repair protein. In addition, 48 out of 80 repair proteins identified in this work were hits in the N-terminally sfGFP-tagged library screening (*Supplementary files 1 and 2*). Screening hits from the N-terminally sfGFP-tagged library can also be a useful resource for further identification of repair proteins.

The screening is unable to identify proteins whose localization remains unaltered in response to SDS treatment. For example, we could not identify Cdc50-Drs2, which accumulates at the damage site induced by a laser (*Yamazaki and Kono, 2022*). A recent study demonstrated the different cellular responses between focal (laser damage) and diffuse (streptolysin-O treatment) PM damage (*Bittel and Jaiswal, 2023*). Some proteins may be overlooked in the screening due to the intrinsic differences between SDS treatment and the laser damage assay. Using different stresses to induce PM damage may identify additional PM repair proteins. In addition, the screening did not identify ESCRT proteins as hits. The tagging of fluorescent proteins occasionally interferes with the functions of the tagged proteins (*Thorn, 2017*). Specifically, the exogenous expression of fluorescent-tagged ESCRT subunits can impair their functions (*Katoh et al., 2003*; *Hoffman et al., 2019*). The screening may have overlooked some of these proteins, possibly including ESCRT proteins.

We performed live-cell imaging of repair proteins and defined accumulation times of repair proteins (*Figure 3—figure supplement 1* and *Figure 2—source data 1*). This dataset provides an overview of repair protein accumulation. However, it should be noted that the accurate relative accumulation timing of repair proteins should be determined by multi-color imaging of repair proteins in the same cells, because of cell-cell variabilities, the GFP-tagging effects on the cells, and the small sample size of the screening. These datasets will form the basis for future hypothesis-driven studies.

## Coordination of polarized exocytosis and CME

We previously showed that formin Bnr1 forms the actin cable at the damage site (*Kono et al., 2012*). Consistent with this, we identified Bnr1-interacting proteins, including Bud6 and Smy1, as repair proteins (*Figure 2—source data 1*). Because actin cables mediate polarized exocytosis and secretory vesicle trafficking (*Pruyne et al., 2004*), these results further support our model that exocytosis delivers repair proteins with TMD at the damage site (*Figure 7*).

We showed that polarized exocytosis and CME occur simultaneously at the damage site between approximately 5 and 35 min after laser damage (*Figure 3C and D*). Given that the resealing of the damaged PM generally occurs within 1 min (*Sønder et al., 2021*), this result implies that polarized exocytosis and CME are involved in the restructuring of the damaged PM rather than the resealing of it. The coupling of endocytosis with polarized exocytosis is observed in multiple cell types, including at the growing tip of budding yeast, the growing tip of pollen tubes, and synapses in neurons (*Gerganova and Martin, 2023*). Polarized exocytosis and CME at the damage site may regulate the PM tensions and amount of lipids and PM proteins in a manner analogous to these cells.

Our results suggest that the activity of CME and polarized exocytosis at the damage site changes over time, with exocytosis predominating within 20 min after laser damage and CME predominating 20 min after laser damage (*Figure 3C and D*). This is consistent with the recent study in human cells

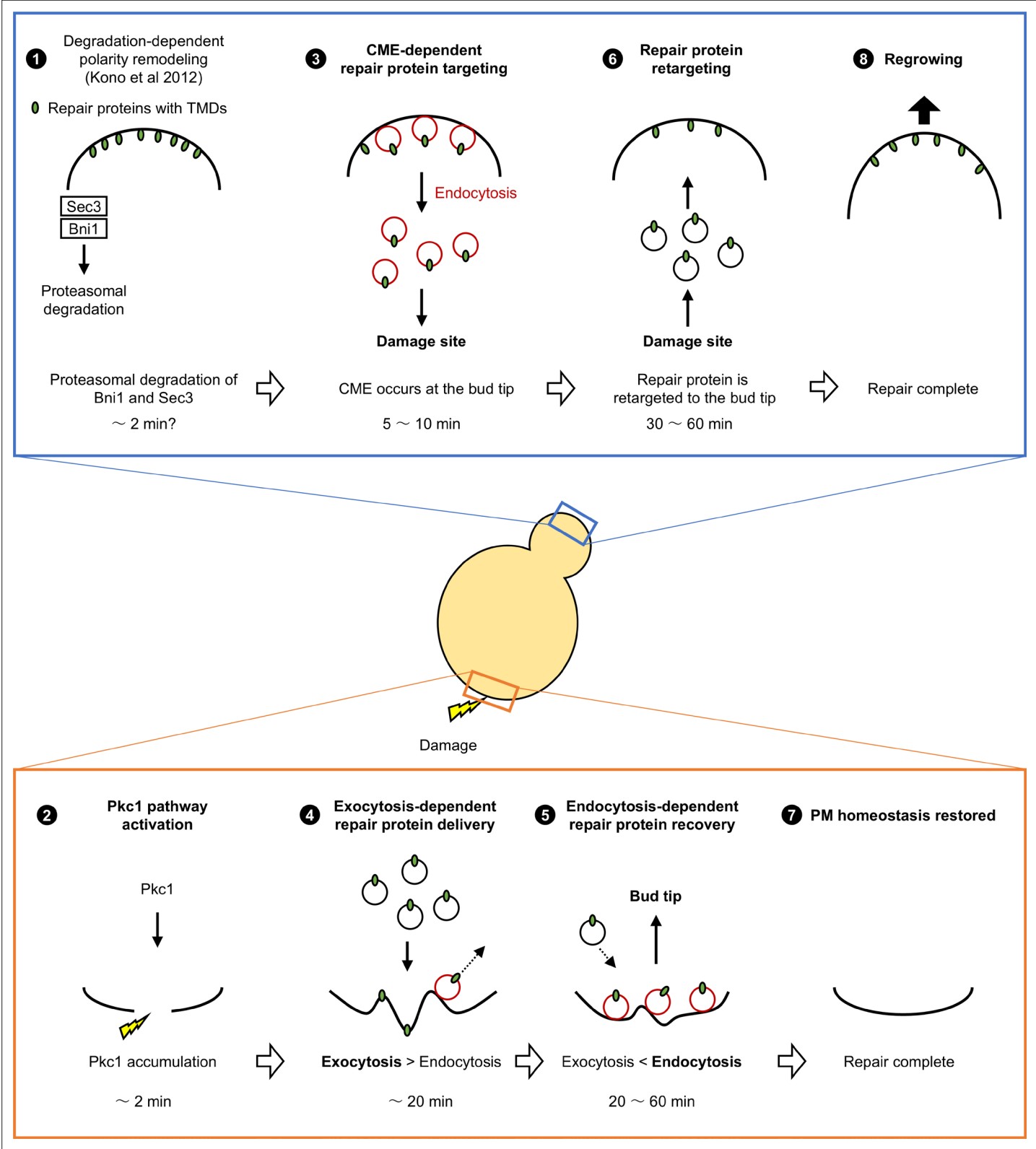

**Figure 7.** Model of spatiotemporal cellular responses to plasma membrane (PM) damage in budding yeast. We showed the hypothetical model of spatiotemporal PM damage responses in budding yeast. First, the degradation of Sec3 and Bni1 resolved the polarity competition between the bud tip and the damage site (**Kono et al., 2012**). Within 10 min after laser damage, clathrin-mediated endocytosis (CME) directs repair proteins with transmembrane domains (TMDs) to the damage site from the bud tip. At the damage site, polarized exocytosis and CME simultaneously occur, with

*Figure 7 continued on next page*

*Figure 7 continued*

exocytosis predominating approximately within 20 min and with CME predominating approximately 20 min after laser damage. CME targets repair proteins with TMDs from the bud tip to the damage site. The endocytosed PM proteins are retargeted to the bud tip again after the PM repair is finished. The retargeted PM proteins may be involved in the resumption of cell growth after PM repair. The numbers represent the temporal order of events.

(*Raj et al., 2024*). A previous study showed that polarized exocytosis activates CME in budding yeast (*Johansen et al., 2016*). Rab GTPase Sec4, which regulates polarized exocytosis, also activates endocytosis by overriding Sla1 inhibition of Las17 (*Johansen et al., 2016*). We could not identify Sec4 as a repair protein candidate because C-terminally GFP-tagged Sec4 is not in the library, probably due to the loss of function of C-terminally tagged Sec4. However, we found that Sec2-GFP, the guanine nucleotide exchange factor (GEF) of Sec4, accumulates at the damage site (*Figure 2—source data 1*). Sec4 and Sec2 are localized to the secretory vesicles (*Elkind et al., 2000*; *Gingras et al., 2022*). Therefore, secretory vesicles that are targeted to the damage site may accumulate Sec2 and Sec4, leading to the activation of CME. The switching of activities between polarized exocytosis and CME may contribute to restoring PM homeostasis after the damage.

Polarized exocytosis at the damage site is inhibited in CME mutants (*Figure 4A and B* and *Figure 4—figure supplement 2A and B*). Given that CME and polarized exocytosis occur simultaneously (*Figure 3C and D*), CME may activate polarized exocytosis at the damage site, such as by regulating the PM tension (*Wang and Galli, 2018*) around the damage site. Moreover, Myo2-sfGFP and Exo70-mNG were partially retained at the bud tip after laser damage in CME mutants (*Figure 4A and B* and *Figure 4—figure supplement 2A and B*). These results raise the possibility that CME is involved in targeting Exo70 and Myo2 from the bud tip. At the bud tip, CME may be involved in the dispersion of Exo70 and Myo2 via upstream regulators, such as Rho3.

## CME functions for PM repair in budding yeast

Previous studies showed that endocytosis actively occurs at the damage site to repair the damaged PM (*Idone et al., 2008*; *Tam et al., 2010*; *Sønder et al., 2021*). Surprisingly, our results suggest that CME occurs not only at the damage site but at the bud tip within 10 min after laser damage. In *rvs167Δ*, in which exocytosis inhibition at the damage site is minimal (*Figure 4B and C*), repair proteins with TMDs at the bud tip fail to accumulate at the damage site (*Figure 5B–F*). These results are consistent with our idea that CME at the bud tip directs repair proteins with TMDs to the damage site (*Figure 7*). Consistent with our results, dysferlin-containing vesicles increase in response to PM damage in muscle cells (*McDade et al., 2014*; *McDade et al., 2021*). Dysferlin has a TMD and functions as a PM repair protein (*McDade et al., 2014*; *McDade et al., 2021*; *Liu et al., 1998*). Delivery of repair proteins with TMDs from non-damaged sites to the damage site by endocytosis may occur in a wide range of eukaryotic species.

We showed that 54–88% of the normalized mNG-Snc1 signal at the bud tip is recovered 50 min after the damage (*Figure 6C and D*). Because Snc1 is a CME cargo and it colocalizes with Ede1-mSc-I at the damage site, it is presumably recovered from the damage site via CME. Previous studies showed that macropinocytosis and CME restructure the damaged PM in human cells (*Sønder et al., 2021*; *Raj et al., 2024*). However, it is not clear whether the cargo proteins are recycled or degraded. We propose that CME terminates the PM damage responses by removing the repair proteins with TMDs from the damage site and resuming the growth by retargeting them to the bud tip (*Figure 7*). Further confirmation of the proposed model requires other experimental strategies, such as the photobleaching and photoconversion of repair proteins, and the use of other repair proteins with TMDs.

Here, we showed spatiotemporal cellular responses after PM damage in budding yeast by large-scale identification of repair proteins and their live-cell imaging. Despite the limitations mentioned above, our datasets provide the first functional catalog for PM repair proteins. Because some of the PM repair proteins identified in this study and CME mechanisms are evolutionarily conserved, this work may serve as a basis for future studies, including those to be conducted in mammalian cells.

# Materials and methods

**Key resources table**

| Reagent type (species) or resource | Designation | Source or reference | Identifiers | Additional information |
|---|---|---|---|---|
| Strain, strain background (*Saccharomyces cerevisiae*) | Budding yeast BY4741 background | *Supplementary file 1* | | |
| Antibody | Anti GFP (Mouse monoclonal) | Roche/Merck | RRID:AB_390913 | WB 1:500 |
| Antibody | Anti Myc (Mouse monoclonal) | Santa Cruz Biotechnology | RRID:AB_627268 | WB 1:250 |
| Antibody | Anti-α-Tubulin (Rat monoclonal) | Bio-Rad | RRID:AB_325005 | WB 1:5000 |
| Software, algorithm | Fij | https://doi.org/10.1038/nmeth.2019 | RRID:SCR_002285 | |

## Media and strains

Standard procedures were used for DNA, *Escherichia coli*, and yeast genetic manipulation. Yeast transformations were performed using the lithium acetate method. A PCR-based procedure was used for gene deletion. The deletion of the expected locus was confirmed by colony PCR (*Longtine et al., 1998*). Yeast cells were cultured in YPD media (1% yeast extract, 2% bacto peptone, and 2% glucose) unless otherwise indicated. SD media (yeast nitrogen base [6.7 g/l] without amino acids, L-adenine (550 mg/l), L-arginine (280 mg/l), L-alanine (280 mg/l), L-asparagine (280 mg/l), L-aspartic acids (280 mg/l), L-cysteine (280 mg/l), glycine (280 mg/l), L-glutamic acids (280 mg/l), L-glutamine (280 mg/l), L-isoleucine (280 mg/l), L-lysine (280 mg/l), L-phenylalanine (280 mg/l), L-proline (280 mg/l), L-serine (280 mg/l), L-threonine (280 mg/l), L-tyrosine (280 mg/l), L-valine (280 mg/l), leucine (530 mg/l), methionine (86 mg/l), histidine (86 mg/l), uracil (22 mg/l), myo-inositol (100 mg/l), and *p*-aminobenzoic acid (3 mg/l) [pH 5.5]) were used for the laser damage assay. Yeast culture was performed at 25°C unless otherwise indicated. The yeast strains, yeast libraries, and plasmids used in this study are listed in *Supplementary files 1–3*.

For the mNG-Snc1 expression experiments in *Figure 6*, we grow cells in SD media containing 2% raffinose instead of glucose overnight. Then, we induced the expression of mNG-Snc1 by adding 3% galactose. After 1 hr of growth in galactose media, the cells were centrifuged at 5000×*g* for 5 min, and the spun-down cells were washed twice with SD media containing 2% glucose. We transferred the spun-down cells to SD media containing 2% glucose. The cells were incubated for at least 3 hr before the laser damage assay.

## GFP screen for SDS damage response

GFP strains were spotted onto the YPD plates from 96-well plates using a pin replicator and incubated at 25°C. After 3 days, the colonies were inoculated into 200 µl of YPD media and incubated overnight to saturation. After mixing, 3 µl of the saturated culture was transferred to 1 ml of YPD media in deep well plates and incubated for 4–5 hr. 500 µl of the cultures were transferred to the new deep well plates. 10 µl of 1% SDS was added to 500 µl of the cultures and incubated at 25°C for 1 hr. The cells were centrifuged at 2000×*g* for 2 min. The cells were fixed with 300 µl of 4% PFA in YPD for 30 min at room temperature. The fixed cells were centrifuged at 2000×*g* for 2 min. Cells were washed twice with PBS and centrifuged at 2000×*g* for 2 min. The supernatant was removed between each wash. Cells were kept at 4°C until imaging. Cells were imaged with an LSM880 confocal microscope using a 20× air objective lens with an Airyscan detector for GFP fluorescence (Ex 488/Em522). Maximum intensity projections of z-stack images are shown.

## Categorization of localization changes in response to SDS treatment

We manually reviewed the acquired images using Zen Blue edition (Zeiss) or Fiji software (*Schindelin et al., 2012*) and compared the fluorescence signals of GFP-tagged proteins in normal and SDS treatment conditions. We identified the proteins whose fluorescence signal pattern changes between the two conditions. We categorized the screening hits based on reported protein localization (*Huh et al.,*

*2003*). We first categorized the bud tip- or the bud neck-localizing proteins as 'From bud tip/neck'. Among the uncategorized remaining screening hits, the actin-localizing proteins were categorized as 'Actin'. Among the remaining uncategorized screening hits, cell periphery-localizing proteins were categorized as 'From cell periphery'. Among the remaining uncategorized screening hits, nucleus-localizing proteins were categorized as 'Nucleus' or 'Nucleus to the cytoplasm'. We categorized Urc1 as 'Nucleus to cytoplasm' because its signal dispersed to the cytoplasm in an SDS treatment condition. Other proteins in the 'Nucleus' showed stronger nucleus localization in the SDS treatment conditions. Among the remaining uncategorized screening hits, we categorized mitochondria-localizing proteins as 'Mitochondria'. Among the remaining uncategorized screening hits, we categorized spindle pole-localizing proteins as 'Spindle pole'. All proteins in the 'Spindle pole' showed stronger dot-like structures in the SDS treatment condition. Among the remaining uncategorized screening hits, we categorized proteins that show puncta or foci in SDS treatment conditions as 'Puncta/foci'. We categorized Gcn2, Ato3, Sun4, Csi2, Sps4, and Ypr089w as 'Puncta/foci dispersed' because they showed weaker foci or puncta signals in SDS treatment conditions. We categorized Tul1, Emp24, Hor7, Scw10, and Ynl019c as 'From vacuole' because they showed decreased vacuole signal in the SDS treatment conditions. We categorized Prm5 as 'to vacuole' because its vacuole signal increases. We categorized Yps1 and Msc1 as 'cytoplasm to ER' because they localize to the ER, and their ER signal increased in SDS treatment conditions. In this study, we make use of the protein localization data from the Saccharomyces Genome Database (SGD) and *Huh et al., 2003*. All results of this screening are shown in *Figure 1—source data 1*.

## Laser damage assay

Yeast cells were grown in SD medium at 25°C until the culture reached an $OD_{600}$ of 0.1–0.4. We diluted the yeast culture and further incubated the yeast cells in SD medium for 3–8 hr at 25°C. We took 1 ml of the culture and centrifuged it at 500×*g* for 5 min to spin the cells down. We took 5–10 µl of the cell suspension and placed it onto SD medium+2.2% agarose bed. A Concanavalin A (ConA) (Nacalai Tesque) coated glass slip was placed on the cells prior to imaging.

Cells were observed with A1R (Nikon). A1R was equipped with a CFI Apochromat TIRF 60×/1.49 oil objective lens (Nikon). GFP, mNG, and sfGFP were excited by a 488 nm laser, and the fluorescence that passed a 525/50 nm band-pass filter was detected with a GaAsP detector. mSc-I was excited by a 561 nm laser, and the fluorescence that passed a 595/50 nm band-pass filter was detected with a GaAsP detector. For the laser damage assay, the 405 nm laser was irradiated to a circle of 0.5 µm diameter in a cell periphery. The laser power was set between 30% and 70%, depending on the fluorescence-tagged proteins. We determined the laser power sufficient for repair protein recruitment at the damage site in control cells without resulting in cell lysis during the experiments. At least ~80% of cells are viable during the experiments, otherwise indicated. Only the cells that survive after laser damage are quantified. To compare the accumulation of repair proteins between different strains, we set the exact same laser power and microscopy.

## Categorization of localization changes in response to laser damage

We categorized proteins that accumulate at the damage site as 'Damage site'. These proteins are defined as repair protein candidates. Among the remaining uncategorized proteins, we categorized cell periphery-localizing proteins as 'PM to cytoplasm'. Among the remaining uncategorized proteins, we categorized Msn2 and Crz1 as 'Nucleus' because they changed localization from the cytoplasm to the nucleus. Among the remaining uncategorized proteins, we categorized Dot6, Dna2, Msc1, and Gcd7 as 'Puncta' because they formed punctate structures in response to laser damage. The repair protein candidates identified in this study are listed in *Figure 2—source data 1*.

## Quantification of fluorescence signal for laser damage assay

The quantification of the fluorescence signal from the laser damage assay was performed as described previously (*Yamazaki and Kono, 2022*). For the large-scale identification of repair protein candidates, we selected the region of interest (ROI) around the whole cell, the damage site, and the bud tip of the cells. We manually move the ROIs as the cell moves so that the ROIs remain at the same position in the cell. We set the ROIs and manually selected them for quantification. The fluorescence intensity at the

damage site and the bud tip was normalized by the fluorescence intensity of a whole cell to minimize the effect of photobleaching during imaging.

We defined the accumulation time as the time when the fluorescence intensity at the damaged site became greater than the fluorescence intensity at the non-damaged site by three times the standard deviation (3×SD) for at least two consecutive time frames. We defined the dispersion time when the fluorescence intensity at the damage site becomes less than that of the non-damaged site plus 3×SD for at least two consecutive time frames after the accumulation time. We defined the retention time as the difference between the accumulation time and the dispersion time. We measured at least three cells. The median values of accumulation times and retention times across replicates for all repair proteins are listed in *Figure 2—source data 1*.

### Growth screening of repair protein knockout mutants

Yeast cultures grown overnight in YPD at 25°C were diluted to $OD_{600}$=0.1. The diluted cultures were spotted onto YPD, YPD+0.01% SDS, and YPD+25 µg/ml CFW plates. After 3 days of incubation at 25°C or 37°C (heat stress), the growth of the YPD plate and other plates was compared. We performed the screening of two independent colonies. The strains were from the yeast deletion collection library (*Winzeler et al., 1999*) except for *smi1Δ*, *las17Δ*, *skg6Δ*, *end3Δ*, *vrp1Δ*, and *sla2Δ*. Only the mutants that showed sensitivity to the stress in two independent colonies were defined as sensitive to the stress. The results of the screening are shown in *Figure 2—source data 1*.

### Spot assay

Yeast cultures grown overnight in YPD at 25°C were diluted to $OD_{600}$=0.1. The fourfold serial dilutions of cultures were spotted onto the indicated plates.

### Immunoblotting

Yeast cells in the early log phase ($OD_{600}$=0.1–0.3) were treated with 0.02% SDS. 5 ml of yeast cells were collected before SDS treatment, 1 hr after SDS treatment, and 2 hr after SDS treatment. Cells were flash-frozen by liquid nitrogen and stored at –80°C. Cells were resuspended in 120 µl of cold lysis buffer (0.25 M NaOH and 1% β-mercaptoethanol) and incubated on ice for 10 min. 20 µl of trichloroacetic acid was added to the lysates. After 10 min of incubation on ice, the lysates were spun down, and the supernatant was discarded. The precipitates were washed with 500 µl ice-cold acetone twice and dried at room temperature. The precipitates were resuspended in SDS polyacrylamide gel electrophoresis (PAGE) sample buffer (63 mM Tris-Cl [pH 6.8], 2% SDS, 1% β-mercaptoethanol, 0.01% bromophenol blue, and 10% glycerol). Immunoblotting was performed with anti-mini c-Myc (Santa Cruz Biotechnology, sc-40), anti-GFP (Roche/Merck, 11814460001), or anti-α-tubulin (Bio-Rad, MCA78G) antibodies.

### Statistical analysis

Dunnett's multiple comparison test, Welch's t-test, and Mann-Whitney U test were performed with Python software (v. 3.11). Gene enrichment analysis and Fisher's exact test were performed with R software (v. 4.2.2).

## Acknowledgements

We thank Dr. Maya Schuldiner (Weizmann Institute) for providing the N' SWAT library. We thank Dr. S Sugiyama for providing yeast strains and technical advice. We also thank the lab members of the Membranology Unit for the discussion and critical reading of the manuscript. We are grateful for the help and support provided by the Imaging Section and Sequence Section of the Core Facilities at Okinawa Institute of Science and Technology Graduate University. We thank Dr. P Barzaghi, Dr. K Koizumi, Dr. S Komoto, and Dr. T Mochizuki for their technical assistance in microscopy experiments. This study was supported by MEXT/JSPS KAKENHI Grant Number 23KJ2138 to YY, JSPS grant-in-aid for scientific research (B) 20H03440, 24K02233, and JST-COI-NEXT JPMJPF2205 to KK.

## Additional information

### Funding

| Funder | Grant reference number | Author |
|---|---|---|
| Japan Society for the Promotion of Science | 23KJ2138 | Yuta Yamazaki |
| Japan Society for the Promotion of Science | 20H03440 | Keiko Kono |
| Japan Society for the Promotion of Science | 24K02233 | Keiko Kono |
| Japan Science and Technology Agency | 10.52926/jpmjpf2205 | Keiko Kono |

The funders had no role in study design, data collection and interpretation, or the decision to submit the work for publication.

### Author contributions

Yuta Yamazaki, Conceptualization, Data curation, Formal analysis, Funding acquisition, Validation, Investigation, Visualization, Methodology, Writing – original draft, Project administration, Writing – review and editing; Keiko Kono, Conceptualization, Supervision, Project administration, Writing – review and editing

### Author ORCIDs

Yuta Yamazaki ⓘ https://orcid.org/0000-0001-5323-6007
Keiko Kono ⓘ https://orcid.org/0000-0003-4057-8790

Reviewer #1 (Public review): https://doi.org/10.7554/eLife.108585.3.sa1
Reviewer #2 (Public review): https://doi.org/10.7554/eLife.108585.3.sa2
Reviewer #3 (Public review): https://doi.org/10.7554/eLife.108585.3.sa3
Author response https://doi.org/10.7554/eLife.108585.3.sa4

## Additional files

### Supplementary files

Supplementary file 1. Yeast strains used in this study.

Supplementary file 2. Yeast libraries used in this study.

Supplementary file 3. Plasmids used in this study.

MDAR checklist

### Data availability

All data needed to evaluate the conclusions in the paper are present in the main text and/or the supplementary materials.

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
