## [Editor Report · eLife Assessment]

This work provides an **important** resource identifying 72 proteins as novel candidates for plasma membrane and/or cell wall damage repair in budding yeast, and describes the temporal coordination of exocytosis and endocytosis during the repair process. The data are **convincing**; however, additional experimental validation will better support the claim that repair proteins shuttle between the bud tip and the damage site.

---

## [Referee Report · Reviewer #1 (Public review)]

Summary:

In this manuscript, Yamazaki et al. conducted multiple microscopy-based GFP localization screens, from which they identified proteins that are associated with PM/cell wall damage stress response. Specifically, the authors identified that bud-localized TMD-containing proteins and endocytotic proteins are associated with PM damage stress. The authors further demonstrated that polarized exocytosis and CME are temporally coupled in response to PM damage, and CME is required for polarized exocytosis and the targeting of TMD-containing proteins to the damage site. From these results, the authors proposed a model that CME delivers TMD-containing repair proteins between the bud tip and the damage site.

Strengths:

Overall, this is a well-written manuscript, and the experiments are overall well-conducted. The authors identified many repair proteins and revealed the temporal coordination of different categories of repair proteins. Furthermore, the authors demonstrated that CME is required for targeting of repair proteins to the damage site, as well as cellular survival in response to stress related to PM/cell wall damage. Although the roles of CME and bud-localized proteins in damage repair are not completely new to the field, this work does have conceptual advances by identifying novel repair proteins and proposing the intriguing model that the repairing cargoes are shuttled between the bud tip and the damaged site through coupled exocytosis and endocytosis.

Weaknesses:

While the results presented in this manuscript are convincing, they might not be sufficient to support some of the authors' claims. Especially in the last two result sessions, the authors claimed CME delivers TMD-containing repair proteins from the bud tip to the damage site. The model is no doubt highly possible based on the date, but caveats still exist. For example, the repair proteins might not be transported from one localization to another localization, but are degraded and re-synthesized. Although the Gal-induced expression system can further support the model to some extent, I think more direct verification (such as FLIP or photo-convertible fluorescence tags to distinguish between pre-existing and newly synthesized proteins) would significantly improve the strength of evidence.

Review on revised version:

The authors addressed most of concerns that were originally raised, primarily by revising the text and figures and expanding the discussion, which improves the clarity of the manuscript. Although the authors did not address my major concern on the shuttling/trafficking model experimentally, I do understand the limitation of resources and time. The authors noted that they planned to do these experiments for their future work, and such studies would be more definitive evaluations for the proposed model. Overall I think this is a very interesting and well-conducted study and I enjoyed reading this manuscript. I look forward to their following research of this study.

---

## [Referee Report · Reviewer #2 (Public review)]

This paper remarkably reveals the identification of plasma membrane repair proteins, revealing spatiotemporal cellular responses to plasma membrane damage. The study highlights a combination of sodium dodecyl sulfate (SDS) and lase for identifying and characterizing proteins involved in plasma membrane (PM) repair in *Saccharomyces cerevisiae*. From 80 PM, repair proteins that were identified, 72 of them were novel proteins. The use of both proteomic and microscopy approaches provided a spatiotemporal coordination of exocytosis and clathrin-mediated endocytosis (CME) during repair. Interestingly, the authors were able to demonstrate that exocytosis dominates early and CME later, with CME also playing an essential role in trafficking transmembrane-domain (TMD) containing repair proteins between the bud tip and the damage site.

Weaknesses/limitations:

- Still, there is a lack of clarity about mentioning Pkc1 as the best characterized repair protein, or why is Pkc1 mentioned only as it is changing the localization?!

- The use of a C-terminal GFP-tagged library for the laser damage assay may have limited the identification of proteins whose localization or function depends on an intact N-terminus. N-terminal regions might contain targeting or regulatory elements; therefore, some relevant repair factors may have been missed. Analysis of endogenously N-terminally tagged strains, at least for selected candidates, could help address this limitation.

- The authors appropriately discuss the limitations of SDS- and laser-induced plasma membrane damage, including the possibility that these approaches may not capture proteins involved in other forms of membrane injury, such as mechanical or osmotic stress.

---

## [Referee Report · Reviewer #3 (Public review)]

Summary:

This work aims to understand how cells repair damage to the plasma membrane (PM). This is important as failure to do so will result in cell lysis and death. Therefore, this is an important fundamental question with broad implications for all eukaryotic cells. Despite this importance, there are relatively few proteins known to contribute to this repair process. This study expands the number of experimentally validated PM from 8 to 80. Further, they use precise laser-induced damage of the PM/cell wall and use live-cell imaging to track the recruitment of repair proteins to these damage sites. They focus on repair proteins that are involved in either exocytosis or clathrin-mediated endocytosis (CME) to understand how these membrane remodeling processes contribute to PM repair. Through these experiments, they find that while exocytosis and CME both occur at the sites of PM damage, exocytosis predominates the early stages of repairs, while CME predominates in the later stages of repairs. Lastly, they propose that CME is responsible for diverting repair proteins localized to the growing bud cell to the site of PM damage.

Strengths:

The manuscript is very well written and the experiments presented flow logically. The use of laser-induced damage and live-cell imaging to validate the proteome-wide screen using SDS induced damage strengthen the role of the identified candidates in PM/cell wall repair.

Comments on revisions:

The authors have very nicely addressed my previous comments and I have no further concerns.

---

## [Author Response]

The following is the authors’ response to the original reviews.

**eLife Assessment**
This work provides an important resource identifying 72 proteins as novel candidates for plasma membrane and/or cell wall damage repair in budding yeast, and describes the temporal coordination of exocytosis and endocytosis during the repair process. The data are convincing; however, additional experimental validation will better support the claim that repair proteins shuttle between the bud tip and the damage site.

We thank the editors and reviewers for their positive assessment of our work and the constructive feedback to improve our manuscript. We agree with the assessment that additional validation of repair protein shuttling between the bud tip and the damage site is required to further support the model.

**Public Reviews:**

**Reviewer #1 (Public review):**
Summary:In this manuscript, Yamazaki et al. conducted multiple microscopy-based GFP localization screens, from which they identified proteins that are associated with PM/cell wall damage stress response. Specifically, the authors identified that budlocalized TMD-containing proteins and endocytotic proteins are associated with PM damage stress. The authors further demonstrated that polarized exocytosis and CME are temporally coupled in response to PM damage, and CME is required for polarized exocytosis and the targeting of TMD-containing proteins to the damage site. From these results, the authors proposed a model that CME delivers TMD-containing repair proteins between the bud tip and the damage site.Strengths:Overall, this is a well-written manuscript, and the experiments are well-conducted. The authors identified many repair proteins and revealed the temporal coordination of different categories of repair proteins. Furthermore, the authors demonstrated that CME is required for targeting of repair proteins to the damage site, as well as cellular survival in response to stress related to PM/cell wall damage. Although the roles of CME and bud-localized proteins in damage repair are not completely new to the field, this work does have conceptual advances by identifying novel repair proteins and proposing the intriguing model that the repairing cargoes are shuttled between the bud tip and the damaged site through coupled exocytosis and endocytosis.Weaknesses:While the results presented in this manuscript are convincing, they might not be sufficient to support some of the authors' claims. Especially in the last two result sessions, the authors claimed CME delivers TMD-containing repair proteins from the bud tip to the damage site. The model is no doubt highly possible based on the data, but caveats still exist. For example, the repair proteins might not be transported from one localization to another localization, but are degraded and resynthesized. Although the Gal-induced expression system can further support the model to some extent, I think more direct verification (such as FLIP or photo-convertible fluorescence tags to distinguish between pre-existing and newly synthesized proteins) would significantly improve the strength of evidence.Major experiment suggestions:(1) The authors may want to provide more direct evidence for "protein shuttling" and for excluding the possibility that proteins at the bud are degraded and synthesized de novo near the damage site. For example, if the authors could use FLIP to bleach budlocalized fluorescent proteins, and the damaged site does not show fluorescent proteins upon laser damage, this will strongly support the authors' model. Alternatively, the authors could use photo-convertible tags (e.g., Dendra) to differentiate between preexisting repair proteins and newly synthesized proteins.

We thank the reviewer for evaluating our work and giving us important feedback. We agree that the FLIP and photo-convertible experiments will further confirm our model. Here, due to time and resource constraints, we decided not to perform this experiment. Instead, we have discussed this limitation in 363-366. Our proposed model of repair protein shuttling should be further tested in our future work.

(2) In line with point 1, the authors used Gal-inducible expression, which supported their model. However, the author may need to show protein abundance in galactose, glucose, and upon PM damage. Western blot would be ideal to show the level of fulllength proteins, or whole-cell fluorescence quantification can also roughly indicate the protein abundance. Otherwise, we cannot assume that the tagged proteins are only expressed when they are growing in galactose-containing media.

Thank you very much for raising the concern and suggesting the important experiments.We agree that the Western blot experiment to confirm the mNG-Snc1 expression in each medium will further strengthen our conclusion. Along with point (1), further investigation of repair protein shuttling between the bud tip and the damage site and the mechanisms underlying it will be an important future direction. As described above, we have discussed this limitation in 363-366.

(3) Similarly, for Myo2 and Exo70 localization in CME mutants (Figure 4), it might be worth doing a western or whole-cell fluorescence quantification to exclude the caveat that CME deficiency might affect protein abundance or synthesis.

We thank the reviewer for suggesting the point. Following the reviewer’s suggestion, we quantified the whole-cell fluorescence of WT and CME mutants and verified that the effect of the CME deletion on the expression levels of Myo2-sfGFP and Exo70-mNG is minimal (Figure S6). We added the description in lines 211-212.

(4) From the authors' model in Figure 7, it looks like the repair proteins contribute to bud growth. Does laser damage to the mother cell prevent bud growth due to the reduction of TMD-containing repair proteins at the bud? If the authors could provide evidence for that, it would further support the model.

Thank you very much for raising the important point. We speculate that the reduction of TMD-containing proteins at the bud by CME is one of the causes of cell growth arrest after PM damage (1). This is because TMD-containing repair proteins at the bud tip, including phospholipid flippases (Dnf1/Dnf2), Snc1, and Dfg5, are involved in polarized cell growth (2-4). This will be an important future direction as well.

(5) Is the PM repair cell-cycle-dependent? For example, would the recruitment of repair proteins to the damage site be impaired when the cells are under alpha-factor arrest?

Thank you for raising this interesting point. Indeed, the senior author Kono previously performed this experiment when she was in David Pellman’s lab. The preliminary results suggest that Pkc1 can be targeted to the damage site, without any impairment, under alpha-factor arrest. A more comprehensive analysis in the future will contribute to concluding the relation between PM repair and the cell cycle.

**Reviewer #2 (Public review):**
This paper remarkably reveals the identification of plasma membrane repair proteins, revealing spatiotemporal cellular responses to plasma membrane damage. The study highlights a combination of sodium dodecyl sulfate (SDS) and lase for identifying and characterizing proteins involved in plasma membrane (PM) repair in *Saccharomyces cerevisiae*. From 80 PM, repair proteins that were identified, 72 of them were novel proteins. The use of both proteomic and microscopy approaches provided a spatiotemporal coordination of exocytosis and clathrin-mediated endocytosis (CME) during repair. Interestingly, the authors were able to demonstrate that exocytosis dominates early and CME later, with CME also playing an essential role in trafficking transmembrane-domain (TMD)containing repair proteins between the bud tip and the damage site.Weaknesses/limitations:(1) Why are the authors saying that Pkc1 is the best characterized repair protein? What is the evidence?

We would like to thank the reviewer for taking his/her time to evaluate our work and for valuable suggestions. We described Pkc1 as “best characterized” because it was the first protein reported to accumulate at the laser damage site in budding yeast (5). However, as the reviewer suggested, we do not have enough evidence to describe Pkc1 as “best characterized”. We therefore used “one of the known repair proteins” to mention Pkc1 in the manuscript (lines 90-91).

(2) It is unclear why the authors decided on the C-terminal GFP-tagged library to continue with the laser damage assay, exclusively the C-terminal GFP-tagged library. Potentially, this could have missed N-terminal tag-dependent localizations and functions and may have excluded functionally important repair proteins

Thank you very much for the comments. We decided to use the C-terminal GFP-tagged library for the laser damage assay because we intended to evaluate the proteins of endogenous expression levels. The N-terminal sfGFP-tagged library is expressed by the NOP1 promoter, while the C-terminal GFP-tagged library is expressed by the endogenous promoters. We clarified these points in lines 114-118. We agree with the reviewer on that we may have missed some portion of repair proteins in the N-terminaldependent localization and functions by this approach. Therefore, in our manuscript, we discussed these limitations in lines 281-289.

(3) The use of SDS and laser damage may bias toward proteins responsive to these specific stresses, potentially missing proteins involved in other forms of plasma membrane injuries, such as mechanical, osmotic, etc.. SDS stress is known to indirectly induce oxidative stress and heat-shock responses.

Thank you very much for raising this point. We agree that the combination of SDS and laser may be biased to identify PM repair proteins. Therefore, in the manuscript, we discussed this point as a limitation of this work in lines 292-298.

(4) It is unclear what the scale bars of Figures 3, 5, and 6 are. These should be included in the figure legend.

We apologize for the missing scale bars. We added them to the legends of the figures in the manuscript.

(5) Figure 4 should be organized to compare WT vs. mutant, which would emphasize the magnitude of impairment.

Thank you for raising this point. Following the suggestion, we updated Figure 4. In the Figure 4, we compared WT vs mutant in the manuscript. We clarified it in the legends in the manuscript.

(6) It would be interesting to expand on possible mechanisms for CME-mediated sorting and retargeting of TMD proteins, including a speculative model.

Thank you very much for this important suggestion. We think it will be very important to characterize the mechanism of CME-mediated TMD protein trafficking between the bud tip and the damage site. In the manuscript, we discussed the possible mechanism for CME activation at the damage site in lines 328-333. We speculate that the activation of the CME may facilitate the retargeting of the TMD proteins from the damage site to the bud tip.

We do not have a model of how CMEs activate at the bud tip to sort and target the TMD proteins to the damage site. One possibility is that the cell cycle arrest after PM damage (1) may affect the localization of CME proteins because the cell cycle affects the localization of some of the CME proteins (6). We will work on the mechanism of repair protein sorting from the bud tip to the damage site in our future work.

**Reviewer #3 (Public review):**
Summary:This work aims to understand how cells repair damage to the plasma membrane (PM). This is important, as failure to do so will result in cell lysis and death. Therefore, this is an important fundamental question with broad implications for all eukaryotic cells. Despite this importance, there are relatively few proteins known to contribute to this repair process. This study expands the number of experimentally validated PM from 8 to 80. Further, they use precise laser-induced damage of the PM/cell wall and use livecell imaging to track the recruitment of repair proteins to these damage sites. They focus on repair proteins that are involved in either exocytosis or clathrin-mediated endocytosis (CME) to understand how these membrane remodeling processes contribute to PM repair. Through these experiments, they find that while exocytosis and CME both occur at the sites of PM damage, exocytosis predominates in the early stages of repairs, while CME predominates in the later stages of repairs. Lastly, they propose that CME is responsible for diverting repair proteins localized to the growing bud cell to the site of PM damage.Strengths:The manuscript is very well written, and the experiments presented flow logically. The use of laser-induced damage and live-cell imaging to validate the proteome-wide screen using SDS-induced damage strengthens the role of the identified candidates in PM/cell wall repair.Weaknesses:(1) Could the authors estimate the fraction of their candidates that are associated with cell wall repair versus plasma membrane repair? Understanding how many of these proteins may be associated with the repair of the cell wall or PM may be useful for thinking about how these results are relevant to systems that do or do not have a cell wall. Perhaps this is already in their GO analysis, but I don't see it mentioned in the manuscript.

We would like to thank the reviewer for taking his/her time to evaluate our work and valuable suggestions. We agree that this is important information to include. Although it may be difficult to completely distinguish the PM repair and cell wall repair proteins, we have identified at least six proteins involved in cell wall synthesis (Flc1, Dfg5, Smi1, Skg1, Tos7, and Chs3). We included this information in lines 142-146 in the manuscript.

(2) Do the authors identify actin cable-associated proteins or formin regulators associated with sites of PM damage? Prior work from the senior author (reference 26) shows that the formin Bnr1 relocalizes to sites of PM damage, so it would be interesting if Bnr1 and its regulators (e.g., Bud14, Smy1, etc) are recruited to these sites as well. These may play a role in directing PM repair proteins (see more below).

Thank you for the suggestion. We identified several Bnr1-interacting proteins, including Bud6, Bil1, and Smy1 (Table S2), although Bnr1 itself was not identified in our screening. This could be attributed to the fact that (1) C-terminal GFP fusion impaired the function of Bnr1, and (2) a single GFP fusion is not sufficient to visualize the weak signal at the damage site. Indeed, in reference 26, 3GFP-Bnr1 (N-terminal 3xGFP fusion) was used.

(3) Do the authors suspect that actin cables play a role in the relocalization of material from the bud tip to PM damage sites? They mention that TMD proteins are secretory vesicle cargo (lines 134-143) and that Myo2 localizes to damage sites. Together, this suggests a possible role for cable-based transport of repair proteins. While this may be the focus of future work, some additional discussion of the role of cables would strengthen their proposed mechanism (steps 3 and 4 in Figure 7).

Thank you very much for the suggestion. We agree that actin cables may play a role in the targeting of vesicles and repair proteins to the damage site. Following the reviewer’s suggestion, we discussed the roles of Bnr1 and actin cables for repair protein trafficking in lines 309-313 in the manuscript.

(4) Lines 248-249: I find the rationale for using an inducible Gal promoter here unclear. Some clarification is needed.

Thank you for raising this point. We clarified this as possible as we could in lines 249255 in the manuscript.

**Recommendations for the authors:**

**Reviewer #1 (Recommendations for the authors):**
(1) The N-terminal GFP collection screen is interesting but seems irrelevant to the rest of the results. The authors discussed that in the discussion part, but it might be worth showing how many hits from the laser damage screen (in Figure 2) overlap with the Nterminal GFP screen hits.

Thank you for the suggestion. We found that 48 out of 80 repair proteins are hits in the N-terminal GFP library (Table S1 and S2). This result suggested that the N-terminal library is also a useful resource for identifying repair proteins. In the manuscript, we discussed it in lines 288-289.

(2) SDS treatment seems a harsh stressor. As the authors mentioned, the overlapped hits from the N- and C-terminal GFP screen might be more general stress factors. Thus, I think Line 84 (the subtitle) might be overclaiming, and the authors might need to tone down the sentence.

Thank you for the suggestion. Following the reviewer’s suggestion, we changed the sentence to “Proteome-scale identification of SDS-responsive proteins” in the manuscript. We believe that the new sentence describes our findings more precisely.

(3) Line 103-106, it does not seem obvious to me that the protein puncta in the cytoplasm are due to endocytosis. The authors might need to provide more experimental evidence for the conclusion, or at least provide more reasoning/references on that aspect (e.g.,several specific protein hits belonging to that group have been shown to be endocytosed).

Thank you very much for raising this point. We agree with the reviewer and deleted the description that these puncta are due to endocytosis in the manuscript.

(4) For Figure 1D and S1C, the authors annotated some of the localization changes clearly, but some are confusing to me. For example," from bud tip/neck" to where? And from where to "Puncta/foci"? A clearer annotation might help the readers to understand the categorization.

Thank you very much for the suggestion. These annotations were defined because it is difficult to conclusively describe the protein localization after SDS treatment. To convincingly identify the destination of the GFP fusion proteins, the dual color imaging of proteins with organelle markers or deep learning-based localization estimation is required. We feel that this might be out of the scope of this work. Therefore, as criteria, we used the localization of protein localization in normal/non-stressed conditions reported in (7) and the Saccharomyces Genome Database (SGD). We clarified this annotation definition in the manuscript (lines 413-436).

(5) For localization in Figure 2C, as I understand, does it refer to6 the "before damage/normal" localization? If so, I think it would be helpful to state that these localizations are based on the untreated/normal conditions in the text.

Yes, it refers to the “before damage/normal localization”. Following the reviewer’s suggestion, we stated that these localizations are based on these conditions in the manuscript (line 130).

(6) The authors mentioned "four classes" in Line 120, but did not mention the "PM to cytoplasm" class in the text. It would be helpful to discuss/speculate why these transporters might contribute to PM damage repair.

Thank you very much for this suggestion. We speculated that these transporters are endocytosed after PM damage because endocytosis of PM proteins contributes to cell adaptation to environmental stress (8). We mentioned it in the manuscript (lines 120-122).

(7) Line 175-180 My understanding of the text is that the signals of Exo70-mNG/Dnf1mNG peak before the Ede1-mSc-I peaks. They occur simultaneously, but their dominating phase are different. It is clearer when looking at the data, but I think the conclusion sentences themselves are confusing to me. The authors might consider rewriting the sentences to make them more straightforward.

Thank you very much for pointing this out. Following the reviewer’s suggestion, we revised the sentence (lines 177-182 in the manuscript).

**Reviewer #2 (Recommendations for the authors):**
It would be interesting to expand on the functional characterization of the 72 novel candidates and explore possible mechanisms for CME-mediated sorting and retargeting of TMD proteins by including a speculative model.

Thank you very much for the comment. We agree that the further characterization of novel repair proteins and exploration of the possible mechanisms for CME-mediated TMD protein sorting and retargeting are truly important. This should be our important future direction.

**Reviewer #3 (Recommendations for the authors):**
The x-axis in Figure 1C is labeled 'Ratio' - what is this a ratio of?

Thank you for raising this point. It is the ratio of the number of proteins associated with a GO term to the total number of proteins in the background. We clarified it in the legend of Figure 1C in the manuscript.

References

(1) K. Kono, A. Al-Zain, L. Schroeder, M. Nakanishi, A. E. Ikui, Plasma membrane/cell wall perturbation activates a novel cell cycle checkpoint during G1 in *Saccharomyces cerevisiae*. Proc Natl Acad Sci U S A 113, 6910-6915 (2016).

(2) A. Das et al., Flippase-mediated phospholipid asymmetry promotes fast Cdc42 recycling in dynamic maintenance of cell polarity. Nat Cell Biol 14, 304-310 (2012).

(3) M. Adnan et al., SNARE Protein Snc1 Is Essential for Vesicle Trafficking, Membrane Fusion and Protein Secretion in Fungi. Cells 12 (2023).

(4) H.-U. Mösch, G. R. Fink, Dissection of Filamentous Growth by Transposon Mutagenesis in *Saccharomyces cerevisiae*. Genetics 145, 671-684 (1997).

(5) K. Kono, Y. Saeki, S. Yoshida, K. Tanaka, D. Pellman, Proteasomal degradation resolves competition between cell polarization and cellular wound healing. Cell 150, 151-164 (2012).

(6) A. Litsios et al., Proteome-scale movements and compartment connectivity during the eukaryotic cell cycle. Cell 187, 1490-1507.e1421 (2024).

(7) W.-K. Huh et al., Global analysis of protein localization in budding yeast.Nature 425, 686-691 (2003).

(8) T. López-Hernández, V. Haucke, T. Maritzen, Endocytosis in the adaptation to cellular stress. Cell Stress 4, 230-247 (2020).